# Analysis of AI Diagnostic Performance Discrepancies Across Medical Imaging Modalities

## Abstract

Artificial intelligence (AI) shows immense promise in medical imaging, yet its diagnostic performance varies significantly across different modalities. This discrepancy is highlighted by the "ultrasound paradox," where AI achieves superior performance on comparatively lower-quality ultrasound images (AUROC 0.94) while struggling with high-resolution, complex modalities like MRI (reported accuracy as low as 0%). This suggests that performance is not dictated by image quality alone but by a complex interplay between the data's intrinsic properties and the structural limitations of current AI architectures. This paper provides a deep-dive analysis of this performance gap by systematically reviewing literature on static, high-contrast (CT, MRI) and dynamic, low-contrast (X-ray, ultrasound) modalities. We investigate the root causes, attributing them to a mismatch between the information type provided by a modality (e.g., spatio-temporal data in ultrasound) and the architectural constraints of dominant AI models like Convolutional Neural Networks (CNNs), such as their limited receptive fields and difficulty in processing temporal features. As a practical solution, we propose a multi-stage "hybrid diagnostic workflow" that strategically combines high-sensitivity AI for initial screening (using X-ray/ultrasound) with high-specificity AI for confirmation (using CT/MRI). This approach aims to optimize overall diagnostic accuracy and clinical efficiency. We conclude that the future of medical AI lies not in a single, universal model but in an integrated, collaborative ecosystem that leverages the unique strengths of different modalities and AI architectures to create robust, clinically-relevant solutions.

## 1 Introduction

Artificial Intelligence (AI) is driving a revolutionary shift in medical imaging, significantly contributing to enhanced diagnostic accuracy and improved clinical workflows. Deep learning algorithms, in particular, demonstrate the ability to recognize complex patterns from large-scale datasets, achieving expert-level diagnostic performance in several domains. A framework developed at UCLA has even shown that deep learning AI can rapidly achieve clinician-level accuracy in complex medical image analysis.

The rapid advancement and practical application of medical imaging AI are evidenced by the fact that approximately 76% of the over 1,000 AI-based medical devices approved by the U.S. FDA are concentrated in radiology. For instance, large-scale studies have shown that AI assistance in breast cancer screening can increase cancer detection rates by 20-30%. In prostate cancer diagnosis, AI has demonstrated the ability to reduce the rate of missed clinically significant lesions from 8% by radiologists to just 1%. These examples underscore AI's contribution to improving diagnostic sensitivity and reading efficiency in real-world clinical settings. While AI has long demonstrated superhuman capabilities in analyzing structured numerical data, such as blood test results, its application to the

Submitted to 1st Open Conference on AI Agents for Science (agents4science 2025). Do not distribute.

unstructured and complex domain of medical imaging reveals a far more nuanced and paradoxical landscape of performance.

However, a notable issue has emerged: the performance of medical AI varies significantly depending on the imaging modality. A systematic review revealed that while ultrasound-based AI models achieved a very high mean Area Under the Receiver Operating Characteristic Curve (AUROC) of 0.94 (95% CI 0.88–1.00), CT and MRI-based models lagged behind at approximately 0.82 (CT: 95% CI 0.78–0.86; MRI: 0.71–0.93). More strikingly, a recent evaluation of the latest ChatGPT-4 vision model reported diagnostic accuracies of around 30% for X-ray images and 40% for CT, but 0% for MRI. This **"ultrasound paradox"**—where the highest performance is observed in a modality with relatively lower image quality—provides compelling evidence that AI performance cannot be predicted by physical image quality alone. It raises a fundamental question about what kind of information AI models learn most effectively and suggests that the performance gap stems not only from the intrinsic properties of the images but also from the structural limitations of current AI architectures.

This study aims to systematically analyze the phenomenon of AI performance discrepancy across imaging modalities, identify its underlying causes, and propose practical solutions. Focusing on the performance differences between static/high-contrast (CT, MRI) and dynamic/low-contrast (X-ray, ultrasound) imaging, we explore the limitations of current AI model architectures and the potential of a hybrid approach to overcome them. Through this analysis, we seek to provide insights that go beyond technical evaluation to inform the future direction of medical AI development and its clinical application strategies.

## 2 AI Performance in Static/High-Contrast Imaging (CT, MRI)

### 2.1 AI Performance in CT Imaging

CT imaging provides favorable conditions for AI model training with its high spatial resolution and excellent tissue contrast. Deep Learning Reconstruction (DLR) techniques have demonstrated superior noise suppression and artifact reduction compared to traditional iterative reconstruction methods, enhancing image quality while reducing radiation exposure [1, 2]

For example, GE Healthcare's 'TrueFidelity' DLR system reconstructs high-quality images with over 50% less radiation, proving effective in detecting liver lesions as small as 0.5 cm. AI's role in lung cancer screening is also noteworthy [3, 4]. Recent studies show that AI systems can automatically track changes in pulmonary nodules across serial CT scans, aiding in the early detection of potentially malignant nodules and assisting clinicians in diagnosis and treatment planning [5, 6]. From an architectural perspective, the 3D volumetric data from CT is advantageous for CNNs to extract hierarchical features layer by layer [7].

However, CNNs' limited local receptive fields make it difficult to capture long-range dependencies, posing a challenge for understanding complex global anatomical relationships [3, 8, 9]. This suggests that Transformer-based models, with their ability to capture global context, could serve as a complementary solution. Indeed, in brain tumor MRI analysis, Vision Transformer (ViT) models have outperformed CNN-based models with over 98% accuracy, highlighting the importance of global information in precision diagnostics [10, 11].

### 2.2 AI Performance and Limitations in MRI Imaging

MRI is an essential modality for the precise diagnosis of conditions like tumors and brain diseases, thanks to its excellent soft-tissue contrast and diverse imaging sequences [12–14]. In specific, well-defined tasks, AI has shown outstanding performance [15, 16]. For instance, a ViT-based model achieved 98.5% accuracy in classifying brain tumors from MRI scans when provided with sufficient data and optimization [17, 18]. Furthermore, AI technology has been developed to reduce the use of gadolinium-based contrast agents by 80-90% while maintaining diagnostic quality, demonstrating the potential to synthesize high-quality images from low-dose contrast scans [19, 20]. This approach is significant for improving patient safety and cost-effectiveness.

Nevertheless, the complex, multi-dimensional data structure of MRI remains a challenge for AI models [21, 22]. The reported 0% diagnostic accuracy of ChatGPT-4 on MRI images underscores

the failure of current general-purpose AI models to comprehend MRI's complexity [23, 24]. MRI data, which includes multiple sequences and 3D spatial information, presents a multi-dimensional problem that is difficult for traditional 2D-centric CNNs to fully capture [25]. This limitation is tied to the architectural constraints of current AI; while CNNs excel at local pattern recognition, they are weak in understanding global correlations and integrating temporal/sequential information [26], which limits their utility in multi-sequence MRI interpretation. Consequently, architectures like Transformers [27], 3D-CNNs [28], or their hybrid models are being proposed as more suitable for MRI analysis [28].

## 2.3 AI Performance Factors in Static/High-Contrast Imaging

The generally stable performance of AI in static/high-contrast imaging like CT and MRI can be attributed to several factors:

**Structural Consistency:** Human anatomical structures appear in relatively predictable and consistent forms in CT and MRI, creating feature maps that are easy for CNNs to learn.

**High Signal-to-Noise Ratio (SNR):** Low noise and clear contrast between tissues make it easier for AI models to distinguish features, enhancing sensitivity even for small lesions.

**Standardized Acquisition Protocols:** The relatively standardized and repeatable examination protocols for CT and MRI ensure consistency in training data, which improves the generalizability of the learned patterns.

**Utilization of 3D Spatial Information:** CT, in particular, provides 3D volumetric data, allowing models like 3D-CNNs to leverage spatial context between adjacent slices to improve diagnostic accuracy.

Thanks to these advantages, the average AUROC for CT-based AI models is reported to be around 0.82 [29], with performance comparable to specialists in tasks like tumor detection and organ segmentation [30]. While MRI performance varies by task, AI has shown expert-level results in fields like neuroimaging [31], though generalizability remains an area for improvement due to the aforementioned structural complexity [32].

# 3 AI Performance in Dynamic/Low-Contrast Imaging (X-ray, Ultrasound)

## 3.1 AI Performance and Limitations in X-ray Imaging

X-ray is the most fundamental and widely used medical imaging modality, serving as a primary examination tool in various fields. Commercial AI-assisted X-ray reading systems are already in use [33], with one independent evaluation of the Rayvolve system reporting a sensitivity of 96.4% and a specificity of 84.4% [34]. This tendency for high sensitivity coupled with somewhat lower specificity is a typical characteristic of X-ray AI [35]. A large multi-center study showed that AI assistance improved the AUC for chest X-ray interpretation by approximately 16% (from 0.759 to 0.88) and reduced reading times.

Key technical challenges for AI in X-ray imaging include:

**Overlapping Structures:** As a 2D projection of 3D information, X-rays suffer from information loss due to overlapping anatomical structures. This can confuse models like CNNs that extract features from local patches and lack global context [36].

**Low Soft-Tissue Contrast:** The low contrast of soft-tissue lesions makes it difficult for models to distinguish the boundaries and shapes of subtle abnormalities [37].

**Variability in Conditions:** X-ray acquisition is subject to high variability from patient positioning, exposure settings, and equipment differences, which can degrade the generalization performance of trained AI models [38].

**Limitations of Local Processing:** Traditional CNNs process images with local filters, making it difficult to capture widespread abnormalities or relationships between distant regions [39]. To address this, research is ongoing into Transformer-based global attention models or adding attention mechanisms to CNNs [40].

## 3.2 Superior AI Performance in Ultrasound Imaging

Surprisingly, AI performance in ultrasound imaging has been reported to surpass that of other modalities. The systematic review previously mentioned found that the average AUROC of 0.94 for ultrasound-based AI was significantly higher than the 0.82 for CT/MRI [41]. This suggests that the real-time nature and diverse information in ultrasound images work to AI's advantage [40]. In breast cancer diagnosis, for example, a deep learning model named DeepBreastCancerNet achieved a remarkable classification accuracy of 99.35% using ultrasound images [42].

Success factors for ultrasound AI include:

**Utilization of Real-Time Dynamic Information:** Ultrasound videos capture temporal changes in organ movement, lesion morphology, and blood flow signals, providing additional information not present in static images.

**Compensating for Operator Dependency:** AI can reduce inter-operator variability by interpreting images based on a consistent, learned standard, thereby raising the overall quality of diagnoses, especially for less experienced practitioners.

**Immediate Feedback and Interaction:** Real-time AI integration can provide immediate alerts for abnormalities during an examination, guiding the operator to perform additional scans or adjust angles.

Common technical challenges across dynamic/low-contrast imaging also exist:

**Difficulty in Learning Spatio-Temporal Features:** Traditional 2D CNNs are ill-equipped to handle the temporal dimension of dynamic videos [43, 44]. Hybrid models like CNN-LSTM are being introduced to address this. For instance, a CNN-LSTM model achieved 97.33% accuracy in predicting bone fracture healing from a series of X-rays, significantly outperforming a pure CNN [45, 46].

**Noise and Artifacts:** Ultrasound's speckle noise and X-ray's scatter and motion blur can degrade AI performance. Pre-processing techniques or noise-robust model architectures are essential [47, 48].

**Lack of Standardization:** The wide variety of equipment, settings, and protocols for ultrasound and X-ray makes it difficult for an AI model optimized in one institution to perform well in another [38]. Domain adaptation and federated learning are being explored to overcome this [49].

In summary, AI performance in X-ray and ultrasound is determined by a combination of the physical limitations of the input data and the structural constraints of current models. The exceptional performance in ultrasound paradoxically highlights these constraints, revealing the potential of AI to leverage temporal and multi-dimensional data when properly equipped.

# 4 A Deeper Look into the Causes of Performance Discrepancy

## 4.1 Hypothesis 1: The Impact of Physical Image Properties on AI Performance

The hypothesis that the physical and technical characteristics of an imaging modality directly impact AI performance is supported by numerous observations [50]. The superiority of static/high-contrast imaging, such as CT and MRI, lies in their high information richness, providing clear anatomical boundaries and relatively low noise [51, 52], which is advantageous for the local pattern learning of CNNs[53].

Conversely, the challenges in dynamic/low-contrast imaging stem from physical limitations [54]. The information loss and low soft-tissue contrast in 2D projected X-rays weaken the signal AI needs to learn from, increasing uncertainty [55]. The 30% accuracy of ChatGPT-4 on X-rays starkly illustrates the negative impact of ambiguous image features [56].

The exceptional performance of ultrasound, however, cannot be explained by traditional image quality metrics alone. Despite its noise and operator dependency, the vast number of frames and diverse acoustic information from real-time scanning appear to benefit AI [57]. This implies that even if physical image quality is lower, AI performance can be high if the quantity and type of information are rich and useful for the model.

## 4.2 Hypothesis 2: The Role of AI Architectural Limitations

The second hypothesis posits that performance discrepancies arise from the inherent limitations of the model architectures themselves. CNNs, the mainstream models in medical imaging AI [58, 59], have structural constraints that negatively affect their performance on certain modalities.

First is the issue of **CNN's local receptive field**. The area of an image a CNN can "see" at once is limited by its filter size and depth [4, 60], making it difficult to understand long-range relationships between distant image regions. This is a disadvantage in images covering large anatomical areas, where global context is crucial. Transformer-based models, with their self-attention mechanisms, have the potential to overcome this limitation by integrating global information [61].

Second is the **inability to learn temporal or dynamic patterns**. CNNs are designed for static 2D images and cannot capture time-varying patterns in videos like ultrasound or longitudinal image series [62]. As mentioned, the significant performance boost from using a CNN-LSTM hybrid for tracking bone healing highlights this deficit [45].

Third is the **complexity of handling multi-dimensional data**. For 3D multi-channel data like MRI, 2D CNNs struggle to extract all necessary volumetric features [63]. While 3D-CNNs exist, they are often limited by high computational costs and data scarcity [64, 65]. Recently, 3D-specific Vision Transformers and the development of large-scale "foundation models" for medical imaging are showing promise in this area.

Recent trends show a move towards **hybrid architectures** like UTNet, Swin-Unet, and ConvFormer, which combine the strengths of CNNs (local detail detection) and Transformers (global context learning) to achieve high performance more efficiently, even in low-data environments [66]

## 4.3 An Integrated Understanding of Performance Discrepancies

Synthesizing these two hypotheses, the performance gap across imaging modalities is best understood as an interaction between the image's characteristics and the AI model's structural properties.

**Information Richness vs. Information Comprehension:** CT/MRI provide physically rich information, but current models may not fully utilize it [67]. Conversely, ultrasound may have less information in terms of resolution but provides it in a form (real-time change) that models can effectively leverage [68].

**Lack of Modality-Specific Architectures:** Most medical AI has been developed using CNNs optimized for static 2D images. This creates a performance deficit for modalities where 3D or temporal information is key (MRI, ultrasound) [69, 70].

**Data and Generalization:** The availability and variability of training data differ by modality [71]. This directly impacts how well a given architecture can realize its potential performance [72, 73].

Ultimately, the physical limitations of an image can be amplified by the constraints of an AI model, or in some cases, complemented by them, as seen with ultrasound. This integrated perspective suggests that the problem should be reframed from "which modality is best?" to "which model is best suited for the unique characteristics of each modality?"

# 5 Discussion

## 5.1 Limitations of Current Research

A review of existing literature reveals several limitations:

**Methodological Bias:** The vast majority (approx. 98%) of medical imaging AI studies are retrospective [74], with a scarcity of prospective studies or randomized controlled trials [75]. This introduces potential bias and may not reflect real-world clinical effectiveness.

**Reporting and Publication Bias:** Many studies claim AI performance is equivalent or superior to clinicians [76, 77], yet less than half (38%) conduct direct comparative evaluations. This suggests a tendency to publish positive results and potentially overstate claims.

**Lack of Standardization and Reproducibility:** Many AI studies fail to adhere to reporting guidelines like TRIPOD [78], omitting crucial details about data pre-processing and model specifics. This raises concerns about the reproducibility and reliability of the findings.

## 5.2 Clinical Significance and Practical Implications

Despite these limitations, the tangible benefits of AI in the clinical setting are undeniable:

**Improved Workflow Efficiency:** AI integration has been shown to reduce image interpretation time by an average of 27.20% and workload by 58.48% [79], alleviating the burden on radiologists.

**Enhanced Diagnostic Accuracy and Consistency:** AI assistance can significantly improve the performance of less experienced physicians, with one study showing a 24% increase in sensitivity [80], helping to standardize the quality of care.

**Patient Safety and Cost Reduction:** AI-driven techniques enable significant reductions in radiation dose (by over 50%) and contrast agent use (by 80-90%), enhancing patient safety while also reducing healthcare costs [81, 82].

## 5.3 A Practical Solution: The Hybrid Workflow (Hypothesis 3)

Based on our analysis, we propose a hybrid diagnostic workflow that strategically combines AI systems with complementary strengths. This multi-stage decision-making process is designed to maximize the advantages of each imaging modality.

**Stage 1 – Broad Screening:** In the initial phase, low-cost, high-sensitivity AI modalities like X-ray or ultrasound are used. The focus is on capturing any potential abnormalities and filtering out the majority of normal cases.

**Stage 2 – Precision Diagnosis:** Cases flagged in Stage 1 proceed to high-resolution, high-specificity modalities like CT or MRI. Here, a second AI system focuses on reducing false positives and accurately characterizing lesions for definitive diagnosis and treatment planning.

**Stage 3 – Integrated Decision:** A clinician makes the final judgment by integrating the results from both stages. This multi-modal ensemble approach has been reported to improve accuracy by over 17% compared to single-modality models [83, 84].

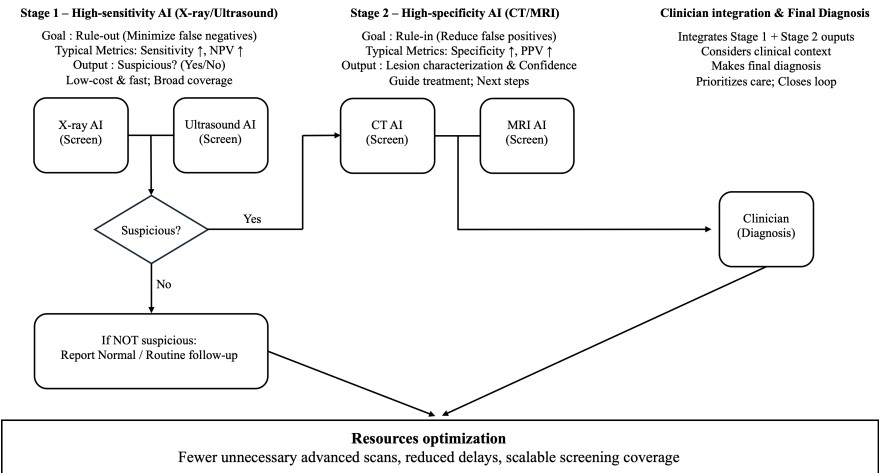

Figure 1: a proposed hybrid diagnostic workflow.
Stage 1 uses high-sensitivity AI (X-ray/ultrasound) for broad screening, and cases flagged as suspicious proceed to Stage 2 for precision diagnosis with high-specificity AI (CT/MRI). A clinician integrates both stages to make the final diagnosis, optimizing resource utilization and reducing diagnostic delays.

## 5.4 Future Research Directions

Future research should focus on enhancing the reliability, efficiency, and applicability of medical imaging AI:

**Explainable AI (XAI):** Developing more intuitive and robust XAI techniques (e.g., SHAP, LIME, Grad-CAM) is crucial to overcome the "black box" nature of deep learning and build clinical trust.

**Foundation Models and Multi-modal AI:** The development of large-scale foundation models pre-trained on millions of medical images could mitigate data scarcity issues. Furthermore, multi-modal AI that integrates imaging with clinical text (e.g., radiology reports) holds great promise for comprehensive clinical decision support.

**Real-time Adaptive Systems:** AI systems that can adapt in real-time to patient-specific characteristics or intra-procedural events are needed. This requires advancements in edge AI and on-device learning.

**Sustainable and Accessible Technology:** Pairing AI with sustainable hardware, such as helium-free MRI and portable ultrasound/X-ray devices, can help bridge global healthcare disparities.

**Data Sharing and Governance:** Privacy-preserving techniques like Federated Learning are essential for collaborative research. Establishing standardized data formats and performance benchmarks is also a key task for the research community and regulatory bodies.

## 6 Conclusion

This study has systematically analyzed the performance discrepancies of AI across different medical imaging modalities, diagnosing their causes and proposing strategic solutions.

**a. Empirical Confirmation of Performance Gaps:** We confirmed that AI performance varies significantly by modality, with ultrasound-based AI showing the highest performance (AUROC 0.94), followed by CT/MRI ($\tilde{0}.82$), while X-ray exhibits greater variability.

**b. A Complex Interplay of Causes:** The performance gap results from a complex interaction between the physical properties of the images and the structural limitations of current AI architectures, particularly the constraints of CNNs in handling global and spatio-temporal information.

**c. The Promise of a Hybrid Workflow:** A hybrid approach that strategically combines the different strengths of modality-specific AIs (high-sensitivity for screening, high-specificity for confirmation) was proposed as a practical and effective solution.

**d. Demonstrated Clinical Value:** AI integration has proven its value by improving workflow efficiency (27% faster interpretation), enhancing diagnostic accuracy (12% sensitivity increase), and improving patient safety (over 50% radiation dose reduction).

The core contribution of this work is the systematic framing of the AI performance gap through a logical progression from **phenomenon → cause → solution**, culminating in the proposal of a practical hybrid workflow. The future of medical AI lies not in perfecting a single model for one modality, but in developing an integrated and collaborative ecosystem where AI, clinicians, and diverse data sources work in concert. Achieving this vision will require continued technological innovation alongside concerted efforts in clinician education, regulatory adaptation, and ethical governance.

## Agents4Science AI Involvement Checklist

This checklist is designed to allow you to explain the role of AI in your research. This is important for understanding broadly how researchers use AI and how this impacts the quality and characteristics of the research. **Do not remove the checklist! Papers not including the checklist will be desk rejected.** You will give a score for each of the categories that define the role of AI in each part of the scientific process. The scores are as follows:

- **[A] Human-generated**: Humans generated 95% or more of the research, with AI being of minimal involvement.
- **[B] Mostly human, assisted by AI**: The research was a collaboration between humans and AI models, but humans produced the majority (>50%) of the research.
- **[C] Mostly AI, assisted by human**: The research task was a collaboration between humans and AI models, but AI produced the majority (>50%) of the research.
- **[D] AI-generated**: AI performed over 95% of the research. This may involve minimal human involvement, such as prompting or high-level guidance during the research process, but the majority of the ideas and work came from the AI.

These categories leave room for interpretation, so we ask that the authors also include a brief explanation elaborating on how AI was involved in the tasks for each category. Please keep your explanation to less than 150 words.

1. **Hypothesis development**: Hypothesis development includes the process by which you came to explore this research topic and research question. This can involve the background research performed by either researchers or by AI. This can also involve whether the idea was proposed by researchers or by AI.

   Answer: **[B]**

   Explanation:The initial ideas for the hypotheses were proposed by human researchers, while the AI evaluated their validity through in-depth research, providing assessments of feasibility along with supporting scholarly papers.

2. **Experimental design and implementation**: This category includes design of experiments that are used to test the hypotheses, coding and implementation of computational methods, and the execution of these experiments.

   Answer: **[B]**

   Explanation: Human authors lack any knowledge in computer science or engineering, rendering them unable to comprehend the experimental designs proposed by the AI. Consequently, the human authors suggested the experimental designs and research methods, which the AI subsequently verified.

3. **Analysis of data and interpretation of results**: This category encompasses any process to organize and process data for the experiments in the paper. It also includes interpretations of the results of the study.

   Answer: **[A]**

   Explanation: As the AI did not directly perform coding or data analysis in this paper, interpretations generated by the AI are not included.

4. **Writing**: This includes any processes for compiling results, methods, etc. into the final paper form. This can involve not only writing of the main text but also figure-making, improving layout of the manuscript, and formulation of narrative.

   Answer: **[C]**

   Explanation: Since some human authors are not native English speakers, AI translation features were extensively utilized. The human authors continually imposed various requirements on the text generated by the AI. For instance, "In our view, our expressions more accurately reflect our intentions than yours. Therefore, we have revised your expressions and sentences."

5. **Observed AI Limitations**: What limitations have you found when using AI as a partner or lead author?

Description: In conducting this research in collaboration with AI, we conclude that the ability to create something from nothing remains a distant goal. Nevertheless, when humans devoid of specialized expertise propose an idea, the AI employs all available means to evaluate it by presenting appropriate rationales. We are confident that this represents a significant advancement in the scientific community, enabling unprecedented innovations through a single idea, without the need for advanced intelligence or knowledge.

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
